# RORβ modulates a gene program that is protective against articular cartilage damage

**Mi Ra Chang, Patrick R. Griffin**⬤*

Department of Molecular Medicine, UF Scripps Biomedical Research, University of Florida, Jupiter, FL, United States of America

* pgriffiin2@ufl.edu

## Abstract

Osteoarthritis (OA) is the most prevalent chronic joint disease which increases in frequency with age eventually impacting most people over the age of 65. OA is the leading cause of disability and impaired mobility, yet the pathogenesis of OA remains unclear. Treatments have focused mainly on pain relief and reducing joint swelling. Currently there are no effective treatments to slow the progression of the disease and to prevent irreversible loss of cartilage. Here we demonstrate that stable expression of RORβ in cultured cells results in alteration of a gene program that is supportive of chondrogenesis and is protective against development of OA. Specifically, we determined that RORβ alters the ratio of expression of the FGF receptors FGFR1 (associated with cartilage destruction) and FGFR3 (associated with cartilage protection). Additionally, ERK1/2-MAPK signaling was suppressed and AKT signaling was enhanced. These results suggest a critical role for RORβ in chondrogenesis and suggest that identification of mechanisms that control the expression of RORβ in chondrocytes could lead to the development of disease modifying therapies for the treatment of OA.

## Introduction

Osteoarthritis (OA) is the most prevalent chronic degenerative joint disease where the risk of disease increases in with age and obesity [1–3]. OA occurs more commonly later in life, after years of mechanical wear and tear on cartilage, a tissue that lines and cushions joints. Most therapies target pain relief and currently there are no effective treatments to slow the progression of the disease. Disease progression eventually results in irreversible loss of cartilage and when articular cartilage is significantly degraded or completely lost, joint replacement surgery is the only option [4]. Although the correlation between aging and the development of OA is not completely understood, it is becoming clear that age-related changes in the musculoskeletal system, combined with mechanical injury and genetic factors all contribute to the pathogenesis of OA [5, 6]. Thus, uncovering the molecular mechanism of joint degeneration requires analysis of cartilage metabolism, chondrocyte senescence, and inflammation [7]. Such studies should lead to the identification of therapeutic targets for treating and preventing OA. Chondrocytes are critical to maintenance of articular cartilage where loss of chondrocytes leads to cartilage damage and this damage is often irreversible. Chondrocyte progenitor cells

**Data Availability Statement:** All relevant data are within the manuscript and its Supporting Information files. The RNA-seq data is deposited in https://www.ncbi.nlm.nih.gov/geo/ GSE208277.

**Funding:** The author(s) received no specific funding for this work.

**Competing interests:** The authors have declared that no competing interests exist.

would be an ideal experimental model for the proposed studies, but markers for such cells are still unclear, and the expression level of several candidate markers such as Notch1 and SOX9 are not consistent on superficial zone (SZ), middle zone (MZ), and deep zone (DZ) of normal tissue and OA tissue [8]. Therefore, in the studies presented here we utilized MG63 cells to overcome these issues. MG63 cells are derived from an established sarcoma cell line and is an osteoblastic model to study bone cell viability, adhesion, and proliferation. Importantly, these cells express the mesenchymal stem cell markers Notch1 and SOX9 and are a well character-ized osteoblast-like cell line which can drive development of OA.

It has been shown that controlled fibroblast growth factor (FGF) signaling is essential for the balance of articular cartilage metabolism, as evidenced by the fact that aberrant FGF signal-ing contributes to progression of OA [9]. FGFR1 signaling triggers upregulation of pro-inflam-matory mediators, matrix metalloproteinases (MMPs), and alters anabolic activities of articular cartilage by inhibition of extracellular matrix (ECM) production and autophagy. Whereas FGFR3 signaling is cartilage protective mainly through the inhibition of pro-inflam-matory mediators and hypertrophic differentiation, as well as reduced MMP expression. For this reason, FGFR1 antagonists and FGFR3 agonists are being pursued as potential therapeutic strategies for OA [10–12].

Nuclear receptors (NRs) represent a druggable superfamily of ligand-dependent transcrip-tion factors. The NR1F subfamily of NRs contains the retinoic acid receptor-related orphan receptors (RORs), which have homology to both the retinoic acid receptors (RARs) and the retinoid X receptors (RXRs) [13]. The ROR subfamily includes three major isoforms, RORα, RORβ, and RORγ. The physiological functions of RORα and RORγ have been well character-ized and they have been shown to play roles in regulation of metabolism and inflammation while RORβ (NR1F2) has been significantly less well studied. RORβ is expressed in regions of the CNS that are involved in processing of sensory information and components of the mam-malian clock, the suprachiasmatic nuclei, the retina, and the pineal gland. RORβ has been shown to play a critical role for the proliferation and differentiation of retinal cells in addition to the maintenance of circadian rhythms [14–16]. The clock gene *BMAL1* (brain and muscle Arnt-like protein 1), a target gene of the RORs, was reported that contributed to the mainte-nance of cartilage homeostasis [17, 18].

Recently, it was shown that RORβ plays a role in osteogenesis by impacting Runx2 target gene expression. Levels of RORβ inversely correlated with osteogenic potential suggesting that suppression of RORβ may drive osteoblastic mineralization. It was shown that RORβ and a subset of RORβ-regulated genes were increased in bone biopsies from post-menopausal women compared to pre-menopausal women suggesting a role for RORβ in human age-related bone loss [19]. Additionally, it was shown that the miR-219a-5p regulates RORβ during osteoblast differentiation and in age-related bone loss [20]. While RORβ$^{-/-}$ mice display bone abnormalities, it was shown that deletion of RORβ was osteoprotective [21].

The main goal of this study was to determine if overexpressing RORβ in an osteoblast-like cell would inhibit the osteoblast phenotype while inducing a chondrogenic phenotype. Here we demonstrate that stable expression of the nuclear receptor RORβ in cultured cells results in alteration of a gene program that is supportive of chondrogenesis and protective against devel-opment of OA. Specifically, RORβ balances the expression of genes implicated in cartilage homeostasis such as altering FGFR signaling towards that favorable for cartilage stability. While it remains unclear how RORβ regulates the balance of FGFR1/3 expression and down-stream signaling, the results presented here suggest RORβ is an important transcription factor involved in the control of a gene program that could prevent articular cartilage damage. Understanding the mechanism of RORβ control of this gene program could lead to the identi-fication of novel therapeutics or drug targets for the prevention and/or treatment of OA.

## Materials and methods

### hRORβ/pLPCX construction and stable expressing transfection

Full length wild type human RORβ (NM_006914.3) was subcloned into the pLPCX retroviral vector (Clontech) using XhoI and NotI restriction enzymes (NEB) with forward primer 5'-CGCGCTCGAGATGCGAGCACAAATTGAAGTGATAC-'3 and reverse primer 5'-GCGGCGGCCGCTCATTTGCAGCCGGTGGCAC-'3. PCR product was obtained with Maxime PCR Premix (i-pfu) (Intron Biotechnologies). pLPCX vector was digested and then dephosphorylated before ligation using the Rapid DNA Dephos & Ligation Kit (Roche). Ampicillin resistance clones were verified using a 5' sequencing primer 5'-AGCTGGTTTAGTGAACCGT-CAGATC-3' and 3' sequencing primer 5'-ACCTACAGGTGGGGTCTTTCATTCCC-3'. Selected clone was linearized with AseI (NEB) restriction digestion and dephosphorylated with Antarctic Phosphatase (NEB) prior usage in transfection.

Human osteoblast-like MG-63 cells (ATCC CRL-1427™) were maintained in EMEM (Eagle's Minimum Essential Medium, ATCC) with 10% heat-inactivated fetal bovine serum (Invitrogen). Linearized hRORβ/pLPCX plasmid was transfected using lipid mediated transfection method (MG-63 transfection kit, Altogen Biosystems). 0.5 ug/ml of Puromycin was added into complete media for selection of stable expressing hRORβ/pLPCX in MG-63 cells. A clone expression high levels of RORβ and a mock-vector clone were used for the following experiments.

### mRNA sequencing analysis

Total RNA was extracted using a Qiagen Kit-74106. Total RNA was quantified using a Qubit 2.0 Fluorometer (Invitrogen, Carlsbad, CA) and run on an Agilent 2100 Bioanalyzer (Agilent Technologies, Santa Clara, CA) for quality assessment. RNA samples of good quality with RNA integrity number (RIN) > 8.0 were further processed. A RNase-free working environment was maintained, and RNase-free tips, Eppendorf tubes, and plates were utilized for the subsequent steps. Messenger RNA was selectively isolated from total RNA (300 ng input) using poly-T oligos attached to magnetic beads and converted to sequence-ready libraries using the TruSeq stranded mRNA sample prep protocol (cat. #: RS-122-2101, Illumina, San Diego, CA). The final libraries were validated on the bioanalyzer DNA chips, and qPCR was quantified using primers that recognize the Illumina adaptors. The libraries were then pooled at equimolar ratios, quantified using qPCR, and loaded onto the NextSeq 500 flow cell at 1.8 pM final concentration. They were sequenced using the high-output, paired-end, 75-bp chemistry. Demultiplexed and quality filtered raw reads (fastq) generated from the NextSeq 500 were trimmed (adaptor sequences) using Flexbar 2.4 and aligned to the mouse reference database (mm9) using TopHat version 2.0.9 (Trapnell, et al.). HTseq-count version 0.6.1 was used to generate gene counts and differential gene expression analysis was performed using Deseq2 (Anders and Huber). We then identified genes that were significantly downregulated or upregulated (adjusted P < 0.05) in each comparison. The RNA-seq data is deposited in https://www.ncbi.nlm.nih.gov/geo/ GSE208277.

### Protein expression analysis

Protein was purified using RIPA lysis and extraction buffer (ThermoFisher Scientific) from MG-63 clones. Halt™ Protease and phosphatase inhibitor cocktail (100X, ThermoFisher Scientific) were added into lysis buffer. To normalize the amount of cell lysate run on SDS-PAGE, BCA (Thermo Fisher Scientific) assay was used. Anti-ROR beta antibody (rabbit monoclonal EPR15552, abcam), anti-phospho ERK1/2 (D13.14.4E, CST), anti-ERK1/2 (L34F12, CST),

anti-phospho AKT (D9E, CST), anti-AKT (40D4, CST), and anti-beta Actin (8H10D10, CST) antibodies were used for primary antibodies. IRDye 680RD and IRDye800CW were used for secondary antibodies. A LI-COR Odyssey system was used for fluorescence imaging. Anti-aggrecan Ab (6-B-4, abcam) was used for primary antibody and FITC-anti mouse Ab was used for secondary antibody. Cell sorting was performed using LSRII (BD Bioscience).

### Quantitative RT-PCR

Total RNA was extracted from RORβ OE/MG63 cells using RNeasy Plus Micro Kit (Qiagen), and the RNA was reverse transcribed using the ABI reverse transcription kit (Applied Biosystems/Thermo Fisher Scientific, Waltham MA). Quantitative PCR was performed with a 7900HT Fast Real Time PCR System (Applied Biosystems) using SYBR green (Roche). A list of primers used for these studies is shown in S1 Table. To measure the effect of IL1β stimulation, 2 nM ILβ (recombinant human IL-1β/IL-1F2, R&D system) was added to WT and RORB OE cells. After 24hr, total RNA was isolated and analyzed by qRT-PCR.

## Results and discussion

### Stable expression of human RORβ in MG63 cells

To determine if overexpressing RORβ in an osteoblast-like cell line would inhibit the osteoblast phenotype while inducing a chondrogenic phenotype MG63 cells were transfected with a pLPCX vector harboring the puromycin-resistance gene and human RORβ cDNA. The transfected cells were selected with puromycin and the expression of RORβ was analyzed by western blotting using an anti-human RORβ specific rabbit EPR1552 mAb with fluorescence imaging using a LI-COR system (Fig 1A). Anti-beta actin (8H10D10) was used as a protein loading control. Relative mRNA expression of hRORβ was determined for clones that survived selection (Fig 1B). One clone demonstrating high expression of RORβ was selected for all subsequent studies. The expression level of RORβ in this clone was confirmed monthly. A control

a.

b.

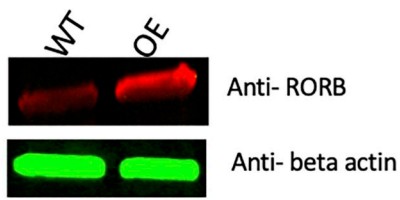
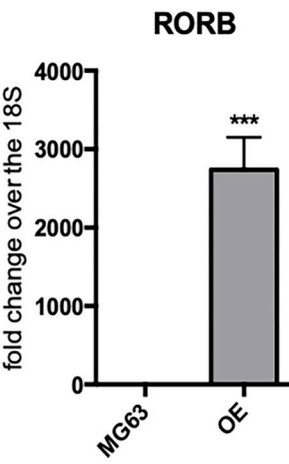

**Fig 1. Selection of MG63 clone which is consistently over expressing hRORβ in MG63 cells.** hRORβ was subcloned into pLPCX vector which was transfected into MG63 cells and selected by 1ug/mL of puromycin. hRORβ was consistently overexpressing in MG63 that was confirmed by **a)** protein expression level and **b)** relative mRNA expression level. WT: mock vector (pLPCX) transfected MG63 cells, OE: Consistently over expressing of hRORβ in MG63 cells.

cell line was generated by transfection of MG63 cells with the pLPCX vector devoid of RORβ and selection with puromycin to generate the wild-type (WT) clone. The WT clone line was used as a control for all studies presented.

## Differential gene expression (DEG) analysis

To investigate the biological function of RORβ in MG63 cells, RORβ OE-MG63 cells (cells overexpressing RORβ) and mock vector transfected MG63 cells (WT clone) were analyzed by mRNAseq. Three independent biological replicates were processed and analyzed. Principle component analysis results and a heatmap of the sample to sample distance are shown in **Fig 2A and 2B**. To investigate molecular pathways altered by RORβ expression, we analyzed log2 fold change of mRNA sequencing data using QIAGEN ingenuity pathway analysis (IPA) software with a P-value cutoff of 0.05. As shown in **Fig 2C**, cells overexpressing RORβ showed alteration in genes involved in signaling pathways associated with OA when compared with WT cells. As further detailed below, the expression profiling results suggest that RORβ may play a protective role in chondrocytes to prevent development of OA.

## RORβ expression suppresses osteoarthritis inducing genes

Aging and chronic inflammation are associated with increase in several critical genes associated with the induction of OA. Matrix metalloproteinases (MMPs) including ADAM

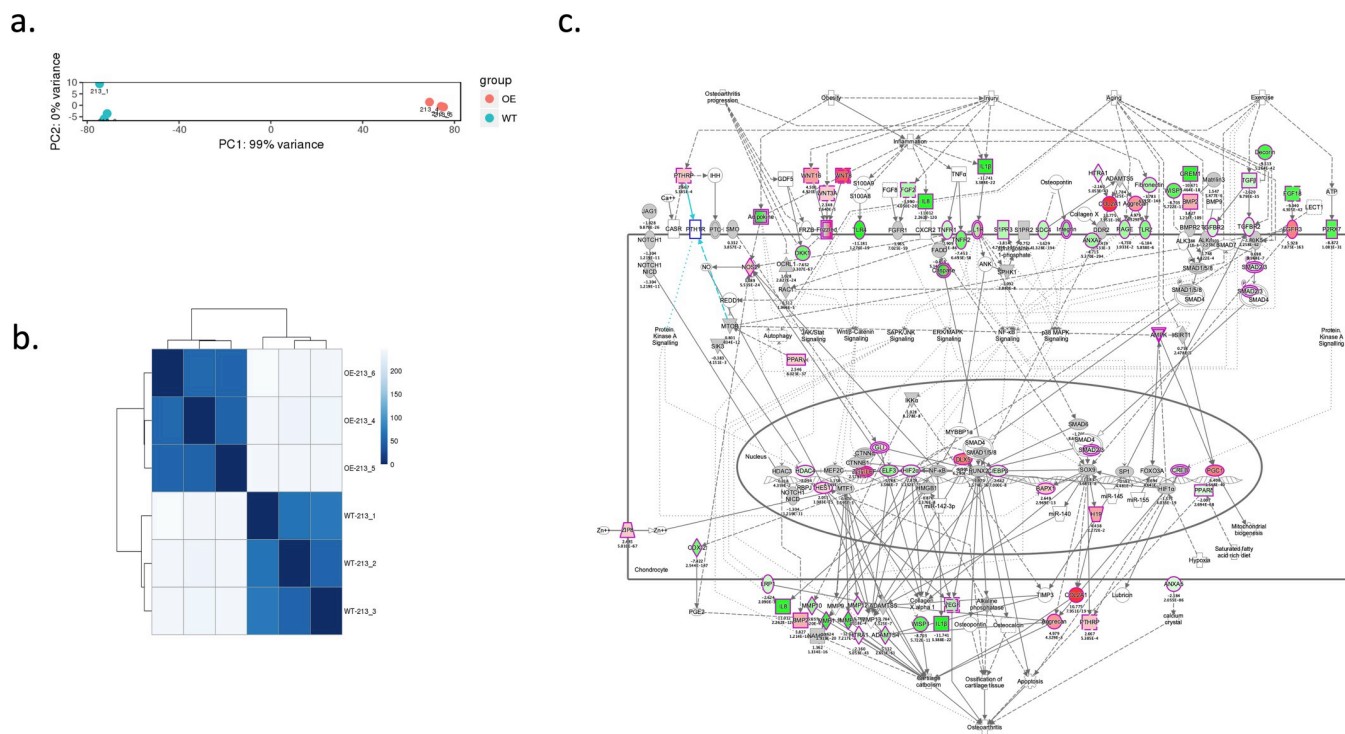

**Fig 2. Differential gene expression sequence. A)** principle component analysis (PCA) of wild type (213–1, 213–2, 213–3) and over expressing RORβ clone (213–4, 213–5, 213–6) in NextSeq 500. Each condition is labeled with a separate color allowing for a visual method for identifying sample outliers. RORβOE cells and WT cells are colored in orange and green, respectively. **B) Heatmap of the sample to sample distance.** The scale is the Euclidean distance between two samples in multi-dimensional space which is also represented in different shades of blue. Each sample is most closely related to itself (with a Euclidian distance of 0) which explains the perfect relatedness (dark blue) along the diagonal. **c)** Suppression of osteoarthritis signaling pathway. mRNAseq data was analyzed using IPA program. The green color of genes represents significantly repressed fold changing genes on OE compared with WT and the red color of genes represent significantly increased fold changing genes on OE compared with WT in OA pathway signaling.

metallopeptidase with thrombospondin type 1 motif 4 (ADAMTS4) and MMP3 and the proin-flammatory cytokine IL6 are well characterized pathogenic genes for OA. Interestingly, increased expression of RORβ in MG63 cells suppressed the expression of these genes (**Fig 3A**) and we demonstrate that the cartilage damaging factor MMP3 was also reduced at the protein level (**Fig 3B**). These results suggest that RORβ may play an inhibitory role to prevent cartilage damage by suppression of chondrocyte degradation by blocking chondrocyte catabolic pathways.

## RORβ induces expression of articular cartilage structural genes

Articular cartilage is composed of collagens, proteoglycans, and non-collagenous proteins. Extracellular matrix proteoglycans and collagens play a critical role in the maintenance of articular cartilage structure by regulating chondrocyte proliferation and promoting cartilage repair [22]. Induction of OA is initiated by mechanical forces and inflammatory events that destabilize the normal coupling of synthesis and degradation of extracellular matrix proteins in both articular cartilage and subchondral bone. In the study presented here, we observed that the expression of the core structural genes in articular cartilage, aggrecan (ACAN) and collagen type II alpha (COL2A1), were increased by overexpressing RORβ in MG63 cells (**Fig 3C**). Additionally, we demonstrate that ACAN protein level was also increased when compared to the WT clone (**Fig 3D**). The levels of ACAN protein in RORβ OE MG63 cells was similar to that observed in TC28a2 cells, which are normal chondrocyte cells. Thus, RORβ inhibits the production of MMPs and increases the expression of extracellular matrix proteins, both of which would be protective to articular cartilage degradation.

## RORβ expression balances FGFR signaling

Fibroblast growth factor (FGF) signaling has a role in growth and homeostasis of joint related cells, including articular chondrocytes, synovial cells, and osteogenic cells and aberrant FGF

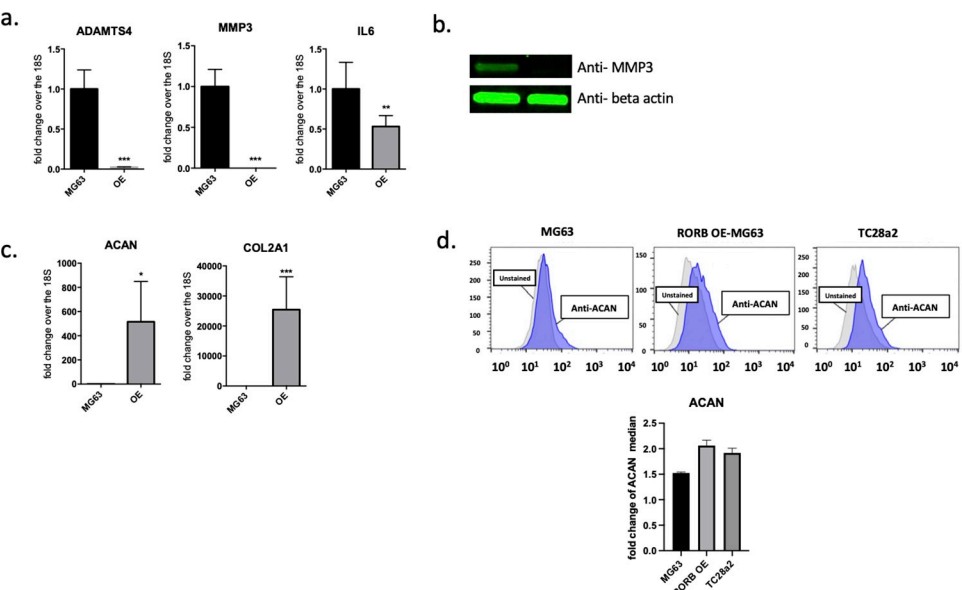

**Fig 3. mRNA expression level for cartilage structure genes and cartilage destruction related genes. a)** Relative mRNA expression for ADAMTS4, MMP3, and IL6 which are direct/indirect cartilage destructive genes and **b)** the protein expression of MMP3 in OE/MG63 clone. **c)** Relative mRNA expression for aggrecan (ACAN) and COL2A1 which is main structure of cartilage maintain and **d)** the protein expression of ACAN on OE/MG63 clone. TC28α2 cells were used for positive control and MG63 cells were used for negative control. Graph with gray color is isotype control.

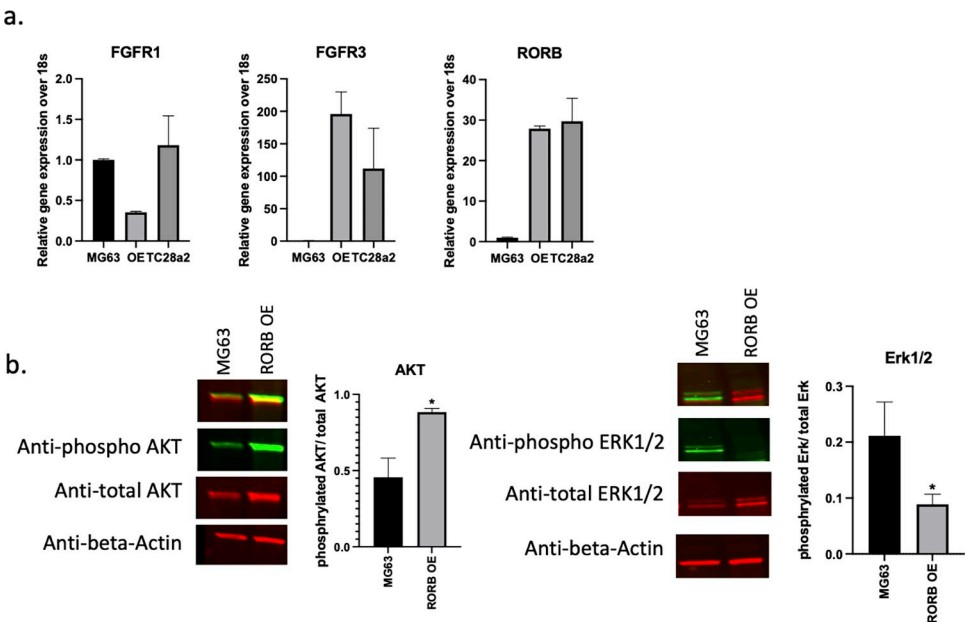

**Fig 4. Reduction of FGFR1 expression and Erk signaling and enhanced FGFR3 expression and Akt signaling in ROR*β* OE cells. a)** Relative mRNA expression of FGFR1 and FGFR3 along with RORβ. **b)** The expression of phosphorylated Erk1/2 and phosphorylated Akt in RORβ over expressing MG63 cells.

signaling contributes to the progression of OA [23]. Genetic inhibition of FGFR1 in knee cartilage attenuates cartilage degradation by the RAF-MEK-ERK and PKCd-p38 pathways [24] and conditional deletion of FGFR3 in mice aggravated DMM-induced cartilage degeneration [25]. In addition, FGF18 attenuates cartilage degradation through FGFR3/PI3K-AKT signaling [26]. As shown in **Fig 4A**, stable expression of RORβ reduced the ratio of FGFR1/FGFR3 towards a protective profile. This alteration of FGFR signaling correlates with changes in phosphorylation of ERK1 (decreased) and AKT (increased) as expected (**Fig 4B**).

## RORβ protects against IL-1β mediated inflammation and inhibits basal protease production

IL-1β is an essential mediator of acute joint inflammation induced by physical injuries and it plays a critical role in cartilage degradation. Induction of IL-1β increases the expression of catabolic matrix enzymes such as MMPs and ADAMTS, as well as the pro-inflammatory cytokine IL-6 which ultimately leads to cartilage matrix degradation [27–29]. Here we demonstrate that stable expression of RORβ suppressed IL-1β induced expression of *IL6*, *MMP3* and *ADAMTS4*. Additionally, expression of RORβ downregulates expression of *MMP3* and *ADAMTS4* as compared to WT control cells (**Fig 5**).

## Conclusions

Development of OA impairs the biomechanical properties of articular cartilage eventually leading to irreversible loss of cartilage resulting in debilitating joint pain and swelling. Owing to the limited regenerative capacity of articular cartilage, advanced surgical techniques have been usually required for repair of damaged cartilage [30, 31]. Risk factors for the development of OA include age, obesity, physical injury, and low-grade systemic inflammation where pro-inflammatory mediators such as IL-6 and TNFα can exacerbate cartilage erosion [32, 33].

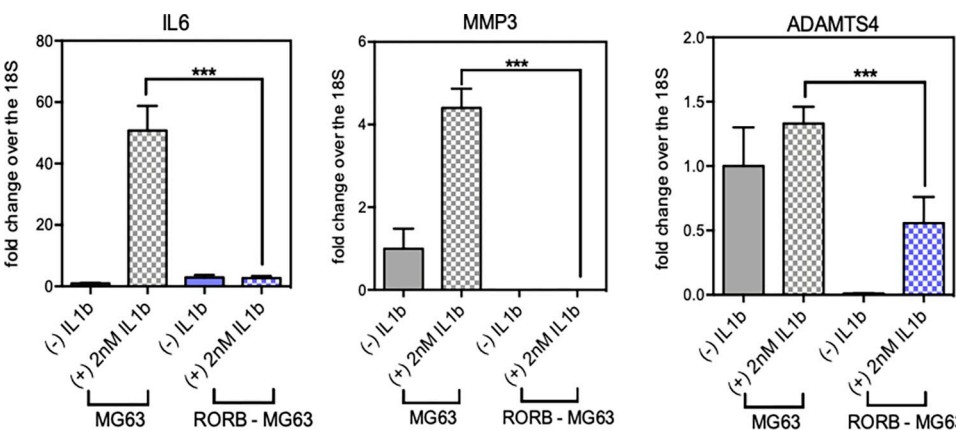

**Fig 5. Suppression of IL1β stimulated genes in hRORβ/MG63 clone.** Inflammatory cytokine IL1β was treated into WT or OE RORβ/MG63 cells for 24 hr with 2nM. The mRNA expression level of IL6, MMP3 and ADAMTS4 was significantly increased by IL1 stimulation but it was protected in RORβ OE cells. The RORβ expression did not affect by IL1β stimulation.

While the pathogenesis of OA remains unclear, hypertrophic chondrocyte differentiation, reduced proliferation, dysregulated apoptosis, combined with the loss of collagen, proteoglycans and cartilage integrity are associated with the development of OA [3].

An important observation in patients with OA is the upregulation of FGFR1 expression which appears to accelerate matrix degradation by inducing the expression of RUNX2 /ELK1 and consequent downregulation of FGFR3 in articular cartilage. Furthermore, pharmacological inhibition of FGFR1 attenuates progression of disease in a mouse model of OA [10] and FGFR3 deficiency in myeloid cells enhances CXCRL12 dependent chemotaxis via CXCR7 (CXC-chemokine receptor 7), thereby leading to the exacerbation of joint destruction [34]. In this study we demonstrate that the nuclear receptor RORβ balances FGFR1/R3 signaling towards that favorable for cartilage stability by suppressing FGFR1 expression and amplifying that of FGFR3. While it remains unclear how RORβ regulates the balance of FGFR1/3 expression, the results presented here suggest RORβ is an important transcription factor controlling a gene program that is protective against articular cartilage damage. Future studies focused on understanding the mechanism of RORβ's control of FGFR1/3 modulation in chondrocytes and murine models of OA could lead to the identification of novel therapeutics and drug targets for the prevention and or treatment of OA.

## Supporting information

**S1 Table. Primer sequences.**
(DOCX)

## Acknowledgments

We thank Ruben Ordonez-Garcia and Anthony Ciesla for generation of the MG63 RORβ stable cell line.

## Author Contributions

**Conceptualization:** Mi Ra Chang, Patrick R. Griffin.

**Data curation:** Mi Ra Chang.

**Formal analysis:** Mi Ra Chang.

**Investigation:** Mi Ra Chang.

**Resources:** Patrick R. Griffin.

**Supervision:** Patrick R. Griffin.

**Validation:** Mi Ra Chang.

**Writing – original draft:** Mi Ra Chang.

**Writing – review & editing:** Mi Ra Chang, Patrick R. Griffin.

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
