## [Decision Letter · Decision Letter 0]

5 Jul 2022

PONE-D-22-12514RORβ modulates a gene program that is protective against articular cartilage damagePLOS ONE

Dear Dr. Griffin,

Thank you for submitting your manuscript to PLOS ONE. After careful consideration, we feel that it has merit but does not fully meet PLOS ONE’s publication criteria as it currently stands. Therefore, we invite you to submit a revised version of the manuscript that addresses the points raised during the review process.

We look forward to receiving your revised manuscript.

Kind regards,

Ewa Tomaszewska, DVM Ph.D

Academic Editor

PLOS ONE

Journal Requirements:

4. Please include a copy of Table 1 which you refer to in your text on page 6.

Additional Editor Comments:

Authors have to justificate the use of MG63 osteoblast-like cells derived from osteosarcoma.

Reviewers' comments:

Reviewer's Responses to Questions

**Comments to the Author**

1. Is the manuscript technically sound, and do the data support the conclusions?

Reviewer #1: No

2. Has the statistical analysis been performed appropriately and rigorously? 

Reviewer #1: I Don't Know

3. Have the authors made all data underlying the findings in their manuscript fully available?

Reviewer #1: No

4. Is the manuscript presented in an intelligible fashion and written in standard English?

Reviewer #1: No

5. Review Comments to the Author

Reviewer #1: This study reports the results of overexpressing retinoic acid recepter-related orphan receptor beta (RORβ) in human osteoblast-like cells (MG63, derived from osteosarcoma), transfecting via pLPCX retroviral vector. A differential effect for RORβ-overexpressing cells versus control was shown using mRNAseq. Several pro-inflammatory genes were reduced in osteoblast cultures overexpressing RORβ. Extracellular matrix genes such as aggrecan and type II collagen were increased. FGFR1 was decreased and FGFR3 was increased. With the assistance of pathway analysis, additional inferences were collected. The ratio of FGFR1/FGFR3 was affected in what is deemed by the authors to be a beneficial response which could balance and mitigate matrix degradation associated with their individual expression. This inference is not clear from the data (Figure 4a,b). Upon induction via IL-1β, catabolic matrix enzymes MMP3, ADAMTS4, and IL-6 were suppressed in RORβ OE culture.

The report is somewhat short. Several important details of methods are omitted, such as no mention of the time course in culture and when assays were conducted. The title and full introduction of the manuscript speaks to chondrocytes and osteoarthritis, and then without mention of justification jumps to utilizing MG63 osteoblast-like cells derived from osteosarcoma. The inferences regarding FGFR ratio do not appear to be supported by the data, warranting full explanation.

The results section includes introductory and methods which would seemingly apply to those sections. The paper does not reference Figure 2a or 2b.

6. PLOS authors have the option to publish the peer review history of their article (what does this mean?). If published, this will include your full peer review and any attached files.

Reviewer #1: No

---

## [Author Response · Author response to Decision Letter 0]

12 Aug 2022

Dear Editor

Thank you for allowing us to submit a revised manuscript. Below you will see that we have thoughtfully and carefully responded to all concerns raised during review. I trust they will find our responses satisfactory.

Sincerely,

Patrick R. Griffin

Editor’s Comments:

Author’s Response: We have made edits to the manuscript to adhere to the PLOS ONE style requirements.

2. In your Data Availability statement, you have not specified where the minimal data set underlying the results described in your manuscript can be found.

Author’s Response: We have added the following statement “All relevant data are within the paper and its Supporting Information files and the RNA-seq data is deposited in https://www.ncbi.nlm.nih.gov/geo/ GSE208277.

4. Please include a copy of Table 1 which you refer to in your text on page 6.

Author’s Response: We apologize for this oversight. Table 1 is now “S1 Table” in the supplemental material. 

Additional Editor Comments:

Authors have to justify the use of MG63 osteoblast-like cells derived from osteosarcoma.

Author’s Response: We have added the following text to justify the use of MG63 osteoblast-like cells. 

“Chondrocyte progenitor cells would be an ideal experimental model for the proposed studies, but markers for such cells are still unclear, and the expression level of several candidate markers such as Notch1 and SOX9 are not consistent on superficial zone (SZ), middle zone (MZ), and deep zone (DZ) of normal tissue and OA tissue [8]. Therefore, in the studies presented here we utilized MG63 cells to overcome these issues. MG63 cells are derived from an established sarcoma cell line and is an osteoblastic model to study bone cell viability, adhesion, and proliferation. Importantly, these cells express the mesenchymal stem cell markers Notch1 and SOX9 and are a well characterized osteoblast-like cell line which can drive development of OA.” 

Reviewer(s)' Comments to Author:

1. Is the manuscript technically sound, and do the data support the conclusions?

Reviewer #1: No

Author’s Response: We address this concern below in response to specific comments from the review.

3. Have the authors made all data underlying the findings in their manuscript fully available?

Reviewer #1: No

Author’s Response: All relevant data are within the paper and its Supporting Information files and the RNA-seq data is deposited in https://www.ncbi.nlm.nih.gov/geo/ GSE208277.

4. Is the manuscript presented in an intelligible fashion and written in standard English?

Reviewer #1: No

Authors’ Response: We have carefully reviewed the manuscript to correct typos and grammatical errors. 

Response to Reviewer #1 

This study reports the results of overexpressing retinoic acid recepter-related orphan receptor beta (RORβ) in human osteoblast-like cells (MG63, derived from osteosarcoma), transfecting via pLPCX retroviral vector. A differential effect for RORβ-overexpressing cells versus control was shown using mRNAseq. Several pro-inflammatory genes were reduced in osteoblast cultures overexpressing RORβ. Extracellular matrix genes such as aggrecan and type II collagen were increased. FGFR1 was decreased and FGFR3 was increased. With the assistance of pathway analysis, additional inferences were collected. The ratio of FGFR1/FGFR3 was affected in what is deemed by the authors to be a beneficial response which could balance and mitigate matrix degradation associated with their individual expression. This inference is not clear from the data (Figure 4a,b). Upon induction via IL-1β, catabolic matrix enzymes MMP3, ADAMTS4, and IL-6 were suppressed in RORβ OE culture.

Author’s response: Our responses are below.

The report is somewhat short. Several important details of methods are omitted, such as no mention of the time course in culture and when assays were conducted. 

Author’s response: We have added additional experimental detail to the methods and figure legends. 

The title and full introduction of the manuscript speaks to chondrocytes and osteoarthritis, and then without mention of justification jumps to utilizing MG63 osteoblast-like cells derived from osteosarcoma. 

Author’s response: We have added the following text to justify the use of MG63 osteoblast-like cells. 

“Chondrocyte progenitor cells would be an ideal experimental model for the proposed studies, but markers for such cells are still unclear, and the expression level of several candidate markers such as Notch1 and SOX9 are not consistent on superficial zone (SZ), middle zone (MZ), and deep zone (DZ) of normal tissue and OA tissue [8]. Therefore, in the studies presented here we utilized MG63 cells to overcome these issues. MG63 cells are derived from an established sarcoma cell line and is an osteoblastic model to study bone cell viability, adhesion, and proliferation. Importantly, these cells express the mesenchymal stem cell markers Notch1 and SOX9 and are a well characterized osteoblast-like cell line which can drive development of OA.” 

The inferences regarding FGFR ratio do not appear to be supported by the data, warranting full explanation.

Author’s response: We have edited the text as follows to clarify the observations related to FGF receptors. 

“We determined that ROR� alters the ratio of expression of the FGF receptors FGFR1 (associated with cartilage destruction) and FGFR3 (associated with cartilage protection). Additionally, ERK1/2-MAPK signaling was suppressed and AKT signaling was enhanced.”

The results section includes introductory and methods which would seemingly apply to those sections. 

Author’s response: Part of this issue is addressed now as the Results section is now “Results and Discussion” to comply with the PLOS ONE style requirements. 

The paper does not reference Figure 2a or 2b.

Author’s response: We apologize for this oversight. We have added text to the document referencing Figures 2a and 2b. 

“Principle component analysis results and a heatmap of the sample to sample distance are shown in Fig. 2a and 2b.”

---

## [Editor Report · Decision Letter 1]

23 Sep 2022

RORβ modulates a gene program that is protective against articular cartilage damage

PONE-D-22-12514R1

Dear Dr. Patrick Robert Griffin,

We’re pleased to inform you that your manuscript has been judged scientifically suitable for publication and will be formally accepted for publication once it meets all outstanding technical requirements.

Kind regards,

Ewa Tomaszewska, DVM Ph.D

Academic Editor

PLOS ONE

Additional Editor Comments (optional):

The authors took into account all comments of the reviewer. Individual ambiguities are explained in the text in the appropriate places. The manuscript can be accepted for publication.
---

## [Editor Report · Acceptance letter]

29 Sep 2022

PONE-D-22-12514R1 

RORβ modulates a gene program that is protective against articular cartilage damage 

Dear Dr. Griffin:

I'm pleased to inform you that your manuscript has been deemed suitable for publication in PLOS ONE. Congratulations! Your manuscript is now with our production department. 

Kind regards, 

on behalf of

Professor Ewa Tomaszewska 

Academic Editor

PLOS ONE